# *ESR1* gene amplification and *MAP3K* mutations are selected during adjuvant endocrine therapies in relapsing Hormone Receptor-positive, HER2-negative breast cancer (HR+ HER2- BC)

Lorenzo Ferrando[1][º], Andrea Vingiani[2,3][º], Anna Garuti[1][º], Claudio Vernieri[4,5], Antonino Belfiore[2], Luca Agnelli[2], Gianpaolo Dagrada[2], Diana Ivanoiu[6], Giuseppina Bonizzi[7], Elisabetta Munzone[8], Luana Lippolis[9], Martina Dameri[10], Francesco Ravera[10], Marco Colleoni[8], Giuseppe Viale[3,7], Luca Magnani[6], Alberto Ballestrero[1,10], Gabriele Zoppoli[1,10][‡], Giancarlo Pruneri[2,3][‡]*

1 IRCCS Ospedale Policlinico San Martino, Genova, Italy, 2 Department of Pathology and Laboratory Medicine, Fondazione IRCCS Istituto Nazionale dei Tumori, Milan, Italy, 3 Department of Oncology and Hemato-Oncology, University of Milan, Milan, Italy, 4 Department of Oncology and Hematology, Fondazione IRCCS Istituto Nazionale dei Tumori, Milan, Italy, 5 IFOM, The FIRC Institute of Molecular Oncology, Milan, Italy, 6 Department of Surgery and Cancer, Imperial College London, London, United Kingdom, 7 Department of Pathology, IEO European Institute of Oncology IRCCS, Milan, Italy, 8 Division of Medical Senology, IEO European Institute of Oncology IRCCS, Milan, Italy, 9 Division of Pathology, Fondazione IRCCS Cà Granda, Ospedale Maggiore Policlinico, Milan, Italy, 10 Department of Internal Medicine (DiMI), University of Genoa, Genova, Italy

º These authors contributed equally to this work.
‡ These Authors contributed equally to the present manuscript.
* giancarlo.pruneri@istitutotumori.mi.it

## Abstract

### Background

Previous studies have provided a comprehensive picture of genomic alterations in primary and metastatic Hormone Receptor (HR)-positive, Human Epidermal growth factor Receptor 2 (HER2)-negative breast cancer (HR+ HER2- BC). However, the evolution of the genomic landscape of HR+ HER2- BC during adjuvant endocrine therapies (ETs) remains poorly investigated.

### Methods and findings

We performed a genomic characterization of surgically resected HR+ HER2- BC patients relapsing during or at the completion of adjuvant ET. Using a customized panel, we comprehensively evaluated gene mutations and copy number variation (CNV) in paired primary and metastatic specimens. After retrieval and quality/quantity check of tumor specimens from an original cohort of 204 cases, 74 matched tumor samples were successfully evaluated for DNA mutations and CNV analysis. Along with previously reported genomic alterations, including *PIK3CA*, *TP53*, *CDH1*, *GATA3* and *ESR1* mutations/deletions, we found that

**Data Availability Statement:** All relevant data are provided within the manuscript and its Supporting Information files.

**Funding:** The study was supported by Fondazione AIRC per la Ricerca sul Cancro, www.airc.it (IG18696 to GP). The funders had no role in study design, data collection and analysis, decision to publish, or preparation of the manuscript.

**Competing interests:** I have read the journal's policy and the authors of this manuscript have the following competing interests: Giancarlo Pruneri reports honoraria from Novartis, Roche, Lilly and Exact Science. Gabriele Zoppoli reports travel grants from Novartis and Pfizer, and reagents for research from Citiva and ThermoFisher Scientific. Andrea Vingiani reports honoraria from Roche and Lilly. Marco Colleoni reports Research Grant from Roche. Elisabtta Munzone reports travel grants from Roche, Pfizer, Lilly and Novartis and reports receiving consultancy fees from Eisai, Exact Sciences, MSD Oncology, Daiichi Sankyo/Astra Zeneca, Pfizer and Seagen. Giuseppe Viale has received grants from Roche/Genentech and Astra Zeneca for his institution; consulting fees from Roche/Genentech, Astra Zeneca, MDS Oncology and Daiichi Sanyko; honoraria for lectures from Roche/Genentech, Astra Zeneca and Daiichi Sanyko; support for attending meetings from Roche/Genentech; and has served on Advisory Boards for Roche/Genentech, Astra Zeneca, Pfizer, MDS Oncology and Novartis. Lorenzo Ferrando, Anna Garuti, Claudio Vernieri, Antonino Belfiore, Luca Agnelli, Gianpaolo Dagrada, Diana Ivanoiu, Giuseppina Bonizzi, Luana Lippolis, Martina Dameri, Francesco Ravera, Luca Magnani and Alberto Ballestrero have no conflict of interest to disclose.

*ESR1* gene amplification (confirmed by FISH) and *MAP3K* mutations were enriched in metastatic lesions as compared to matched primary tumors. These alterations were exclusively found in patients treated with adjuvant aromatase inhibitors or LHRH analogs plus tamoxifen, but not in patients treated with tamoxifen alone. Patients with tumors bearing *MAP3K* mutations in metastatic lesions had significantly worse distant relapse-free survival (hazard ratio [HR] 3.4, 95% CI 1.52–7.70, p value 0.003) and worse overall survival (HR 5.2, 95% CI 2.10–12.8, p-value < 0.001) independently of other clinically relevant patient- and tumor-related variables.

## Conclusions

*ESR1* amplification and activating *MAP3K* mutations are potential drivers of acquired resistance to adjuvant ETs employing estrogen deprivation in HR+ HER2- BC. *MAP3K* mutations are associated with worse prognosis in patients with metastatic disease.

## Author summary

Breast cancer is the most frequently diagnosed cancer and represents the leading cause of cancer-related death in women. Hormone receptor positive tumors account for 70–80% of all breast cancers. They are characterized by estrogen dependent growth, and are routinely treated by endocrine therapy, aiming at blocking estrogen receptor (e.g., tamoxifen) or inhibiting the production of estrogen (aromatase inhibitors and LHRH analogues). Unfortunately, a significant proportion of patients develops endocrine resistance, ultimately leading to tumor recurrence. In this study, we analyzed a cohort of 74 hormone receptor positive breast cancer patients by performing a deep molecular characterization of treatment-naïve primary tumor samples and their matched metastatic localizations, to highlight putative mechanisms of endocrine resistance. Along with expected acquired molecular alterations, including mutation in *ESR1* gene, that encodes for estrogen receptor, we found that an increase of the number of copies of the *ESR1* gene (amplification) and mutations in *MAP3K* are significantly enriched in relapsing tumors, thus expanding the spectrum of known endocrine therapy resistance mechanisms. Interestingly, we found that patients with *MAP3K* mutations were associated with a worse prognosis.

## Introduction

Recent works aiming at a broad biological characterization of metastatic Hormone Receptor-positive, Human Epidermal growth factor Receptor 2-negative breast cancer (HR+ HER2-BC) have revealed a complex and heterogeneous genomic landscape [1–4]. In the largest clinical series published so far, several alterations in oncogenes or tumor suppressor genes (TSGs) occurring more commonly in metastatic HR+ HER2-BC (mBC) than in early BC (eBC) specimens were reported [2–4]: the most frequent oncogenic mutations involved *ESR1*, *ERBB2* and *FGFR4* genes, while *TP53*, *RB1*, *ATR*, *FAT1* and *ARID1* were the most commonly altered TSGs. *ESR1* mutations leading to constitutive activation of Estrogen Receptor alpha (ERα) were selected by pharmacological treatments leading to reduction of estrogen levels in peripheral blood and in tumor microenvironment, such as LHRH analogues (LHRHa) and aromatase inhibitors (AIs) [5,6].

Although the aforementioned analyses provided a broad picture of the most common genomic alterations occurring in metastatic HR+ HER2- BC specimens, they mostly relied on a small number of paired eBC and mBC samples [3], or they did not report patient therapy [4], thus hindering a full understanding of the interaction between specific treatments and dynamic changes of tumor mutational profiles.

Herein, we report the results of a retrospective study in which we used a customized Next Generation Sequencing (NGS) panel of 134 BC-related genes to investigate the genomic evolution of HR+ HER2- BC in patients treated with adjuvant ET after curative surgery and then undergoing disease relapse during/after adjuvant ET. We show that *ESR1* amplifications and *MAP3K* mutations are selected during ETs employing estrogen deprivation, and that *MAP3K* alterations are associated with higher risk of tumor relapse and worse patient overall survival.

## Methods

### Ethics statement

The study was approved by the Ethics Committee of European Institute of Oncology (IEO) (ID18696). All patients enrolled in this study signed an informed consent for the use of their clinical and genomic data for research purposes.

### Study cohort and primary-metastatic matched pairs

Patients included in this study were selected from a case series of the European Institute of Oncology (IEO) based on the following inclusion criteria: 1) Hormone Receptor-positive (HR +) BC, as defined as ≥ 1% expression of estrogen receptor (ER) and/or progesterone receptor (PgR) based on immunohistochemistry (IHC); 2) Human Epidermal growth factor Receptor 2 (HER2)-negative disease, as defined as an IHC score of 0, 1+, or 2+ in the absence of HER2 gene amplification at ISH evaluation (ASCO/CAP guidelines); 3) availability of clinical data regarding patient age, tumor stage, tumor grade, Ki-67 labelling index in tumor cells at diagnosis, menopausal status, adjuvant chemotherapy (if any), adjuvant ET, time of tumor relapse, overall survival; 4) availability of FFPE specimens of matched primary and metastatic tumor samples for genomic evaluations; 5) availability of a source of germline DNA; 6) tumor relapse occurring at least six months after BC diagnosis (to exclude patients with *de novo* metastatic disease); 7) tumor diagnosis after year 2000; 8) cellularity of primary and/or metastatic tumor specimens equal to, or higher than 10%.

### Pathology and *ESR1* amplification FISH assessment

All primary and metastatic tumor samples from patients included in the study were retrieved from IEO pathology archives and reviewed by two expert pathologists (G.P., A.V.) for diagnosis confirmation. Tumor type, grade, ER, PgR and HER2 status, Ki-67 labeling index, occurrence of peritumoral vascular invasion, nodal status and type of surgery were defined [7–10] and recorded. Somatic copy number alterations (sCNA) of *ESR1* genes were assessed by FISH, using Bacterial Artificial Chromosome (BAC) clones obtained from C.H.O.R.I. (bac-pac resources, Children's Hospital Oakland Research Institute, California, US), labeled by means of nick translation (Nick Translation Reagent Kit, Abbott Molecular, Chicago, Illinois, US), and validated on normal metaphase spreads. FISH evaluations were performed on FFPE sections using standard protocols. The detailed description of the pathology assessment is described in the supplementary methods (S1 Star Methods).

## Targeted gene sequencing panel design

To cover the highest number of BC-related genomic aberrations, we designed a custom, amplicon-based, 6,812-amplicon targeted gene sequencing (TGS) genomic panel spanning 530.65 Kbps of the human exome. The panel was optimized by ThermoFisher Scientific for automated fluidic handling with two primer pools, using an Ion Chef robot, and amplicons were designed to be used with FFPE degraded material. Targets were selected from publicly available BC-omics datasets [1,11–17] at the time of TGS panel customization. The details of the design of the targeted panel are reported in the supplementary methods (S1 Star Methods).

## DNA extraction, library preparation and sequencing

Archival FFPE sample were manually macrodissected and tumor cellularity of each sample was recorded. DNA was extracted using the Maxwell RSC DNA FFPE Kit with a Maxwell RSC Instrument (Promega Corporation, Madison, USA). The concentration and purity of DNA samples was measured using the Qubit dsDNA HS Assay Kit (Invitrogen) on a Qubit 2.0 Fluorometer (Invitrogen). Libraries were generated using 10 ng input DNA using ThermoFisher Scientific Ion Chef. Multiplexed libraries were then sequenced using a Ion GeneStudio S5 Plus System. TGS reads were aligned against GRCh38 reference. Filtering cutoffs for proceeding to downstream analysis were as follows: for files derived from normal tissues Q20 < 150,000,000 bases, number of mapped reads < 1,500,000, coverage depth < 100x, percent on target at 20x < 90%; for files originating from cancer samples Q20 < 700,000,000 bases, number of mapped reads < 7,000,000, coverage depth < 1000x, percent on target at 100x < 80%. Such values were chosen as the excess approximation of the 10th percentile of each considered measure for all the initially sequenced files.

## Mutation, copy calling, and variant prioritization

Mutation calling was then performed using the default settings of the ThermoFisher somatic-germline paired mutation pipeline. Sequencing errors were filtered using the proprietary algorithm by Thermo Fisher SVB. Variants obtained were independently validated with alternative pipelines and prioritized, as described in supplementary methods (S1 Star Methods). sCNA were computed using ONCOCNV [18], optimized for amplicon-based TGS data. Ploidy and absolute copy number for each sample were estimated using ABSOLUTE [19] (v2.0). Gene Ontology (GO) term enrichment was performed fitting a generalized linear model. GO terms were annotated using AnnotationDbi [20] R package.

## Statistical analysis

Enrichment of frequencies of altered genes (mutations and sCNA) between primary and metastatic samples was evaluated by using a mixed-effect model (*lme4* R package [21]). This allowed us to adjust for the pairing covariate of samples belonging to the same patient. Confidence intervals, based on the Clopper-Pearson exact confidence interval, were estimated using the *exactci* function from *PropCIs* package [22]. *P*-values were calculated using McNemar's test, specifically designed for paired data, and adjusted for false discovery rate using the Benjamini-Hochberg approach. Identification of modules of mutually exclusive gene triplets was performed using CoMEt [23]. The analysis included both mutations and sCNA. Associations between specific genomic alterations and type of adjuvant ET were studied by fitting a generalized linear model with binomial family, brglmFit method and MPL_Jeffreys to adjust the model for matching covariate. Survival analysis was carried out using Cox proportional hazards regression model. The impact of individual genomic alterations on distant relapse-free

survival (DRFS) and overall survival (OS) was adjusted for the following covariates: T size, N status, Ki-67 labeling index, tumor grade and menopausal status. Gene Ontology terms associated with specific gene mutations were selected for reporting if showing a p-value for generalized linear regression with the number of acquired mutations < 0.05 and for Cox regression < 0.1 (unadjusted for exploratory purposes), and if present in at least 5% of mBC cases. All the analyses were performed with R version 3.5.1.

## Results

### Patient selection and sequencing statistics

We evaluated 204 patients potentially fulfilling the inclusion criteria. Of these, 19 patients were excluded because eBC and/or mBC tumor specimens for genomic analyses could not be retrieved. Another 61 patients were excluded because of the lack of a reliable source of germline DNA, or due to subthreshold quality of extracted DNA from at least one of the three matched samples (eBC, mBC, germline), as described in the Methods, or because invasive BC cells in FFPE tumor blocks or biopsies were exhausted upon centralized revision. Finally, another 50 patients were excluded because any of the three matched samples did not meet QC sequencing criteria described in the Methods (**S1 Fig** and **S1 Star Methods**). We finally included 74 patients with high-quality eBC, mBC and normal tissue specimen available, and which met all the established criteria for genomic evaluation. For primary (eBC) samples entering our final analyses, median sequencing depth was 2,728x (IQR: 2,407–3,064), with 94.3% of the targeted regions covered at 500x or more (IQR: 91.9–95.8). In metastatic (mBC) samples, median sequencing depth was 2,744x (IQR: 2,478–3,215), with 90.9% of the targeted regions covered at 500x or more (IQR: 85.3–94.4). Finally, in normal samples, median sequencing depth was 838x (IQR: 668–1,086), with 97.2% of the targeted regions covered at 100x or more (IQR: 94.5–97.8).

### Patient characteristics

Patient and tumor-related characteristics, as well as clinical outcomes of 74 patients included in the present analysis, are summarized in **Table 1**. Median patient age at eBC diagnosis was 45.5 years, with most patients being premenopausal (67.6%). The most common histologic type was ductal (78.4%), followed by lobular (16.2%). The majority of patients had a "luminal B-like" phenotype, with 62.2% of cases showing a Ki-67 labelling index equal to or greater than 20%, and with 75% of tumors with histological grade 3 (G3). Regarding adjuvant therapies, 63.5% of patients received (neo)adjuvant chemotherapy, mostly consisting of anthracycline-containing regimens (43.2% of the whole cohort), followed by adjuvant ET, while 36.5% of patients received adjuvant ET only. As for adjuvant ETs, 54.1% of patients received ovarian suppression (through LHRHa) with TAM or AIs, 28.4% received AIs alone, and 14.9% received TAM alone. In three patients enrolled in randomized clinical trial, we were unable to retrieve information about the type of ET prescribed.

In the study cohort, median DRFS was 74.4 months (95% CI: 62.0–97.7), while median OS was 117.9 months (95% CI: 99.7–146.2). Overall, our study cohort was enriched in aggressive HR+ HER2- BCs, as suggested by the high proportion of young and premenopausal women included, as well as by the high frequency of high-grade, luminal B-like cancers. This enrichment likely results from the fact that for the aims of this work we purposely selected patients relapsing after curative surgery. No statistically significant differences in terms of clinico-pathological characteristics were found between the initially screened 204 patients and the 74 patients who were finally included in this analysis (**S3 Table**).

**Table 1. Demographic and clinical characteristics of the study cohort.**

| Age at Diagnosis (median, IQR) | 45.5 (40–53) |
|---|---|
| **Menopausal status** | |
| Pre | 50 (68%) |
| Post | 24 (32%) |
| **Histological subtype** | |
| Ductal invasive carcinoma | 58 (78%) |
| Lobular invasive carcinoma | 12 (16%) |
| Mixed invasive carcinoma | 4 (6%) |
| **Grade** | |
| x/n.a | 16 (22%) |
| 1 | 2 (3%) |
| 2 | 29 (39%) |
| 3 | 27 (36%) |
| **T size** | |
| pT1 | 26 (35%) |
| pT2> | 48 (65%) |
| **Lynphonode Status** | |
| Negative | 17 (23%) |
| Positive | 57 (73%) |
| **Ki67** | |
| Luminal A-like | 27 (36%) |
| Luminal B-like | 46 (62%) |
| n.a | 1 (1%) |
| **Adjuvant Chemotherapy regimen** | |
| Anthracycline containing | 32 (43%) |
| No chemotherapy | 27 (37%) |
| Other | 15 (20%) |
| **Adjuvant endocrine treatment** | |
| Aromatase inhibitors (AI) | 12 (16%) |
| LHRH + AI | 3 (4%) |
| Tamoxifen followed by AI | 5 (7%) |
| Tamoxifen (TAM) | 11 (15%) |
| LHRH + TAM | 40 (54%) |
| Random Clinical Trial | 3 (4%) |
| **Relapse-free interval (mm, 95% CI)** | 74.5 (62.0–97.7) |

## Matched genomic analysis reveals novel putative drivers of acquired resistance to ET

The genomic landscape of 74 matched HR+ HER2- BC specimens, including point mutations, indels and sCNA in 134 BC-related genes is depicted in **Fig 1**. Overall, the incidence of the most common mutations found in our dataset matches that of previously reported analyses[2-4]. In particular, *PIK3CA*, *TP53*, and *CDH1* were the most frequently altered cancer genes, with a prevalence of 39%, 31%, and 24%, respectively (**Fig 1A**). Among commonly altered genes, *PIK3CA*, *CCND1* and the FGF genes' cluster (which, together with *CCND1*, is physically co-located on the same chromosome cytoband, 11q13.3) did not gain *de novo* mutations in mBC specimens as compared to their matched eBC specimens. By contrast, we found an enrichment in alterations of *TP53* (8%), *CDH1* (11%), *ESR1* (16%), *FGFR1* (5.4%), and *PTEN* (4%) genes

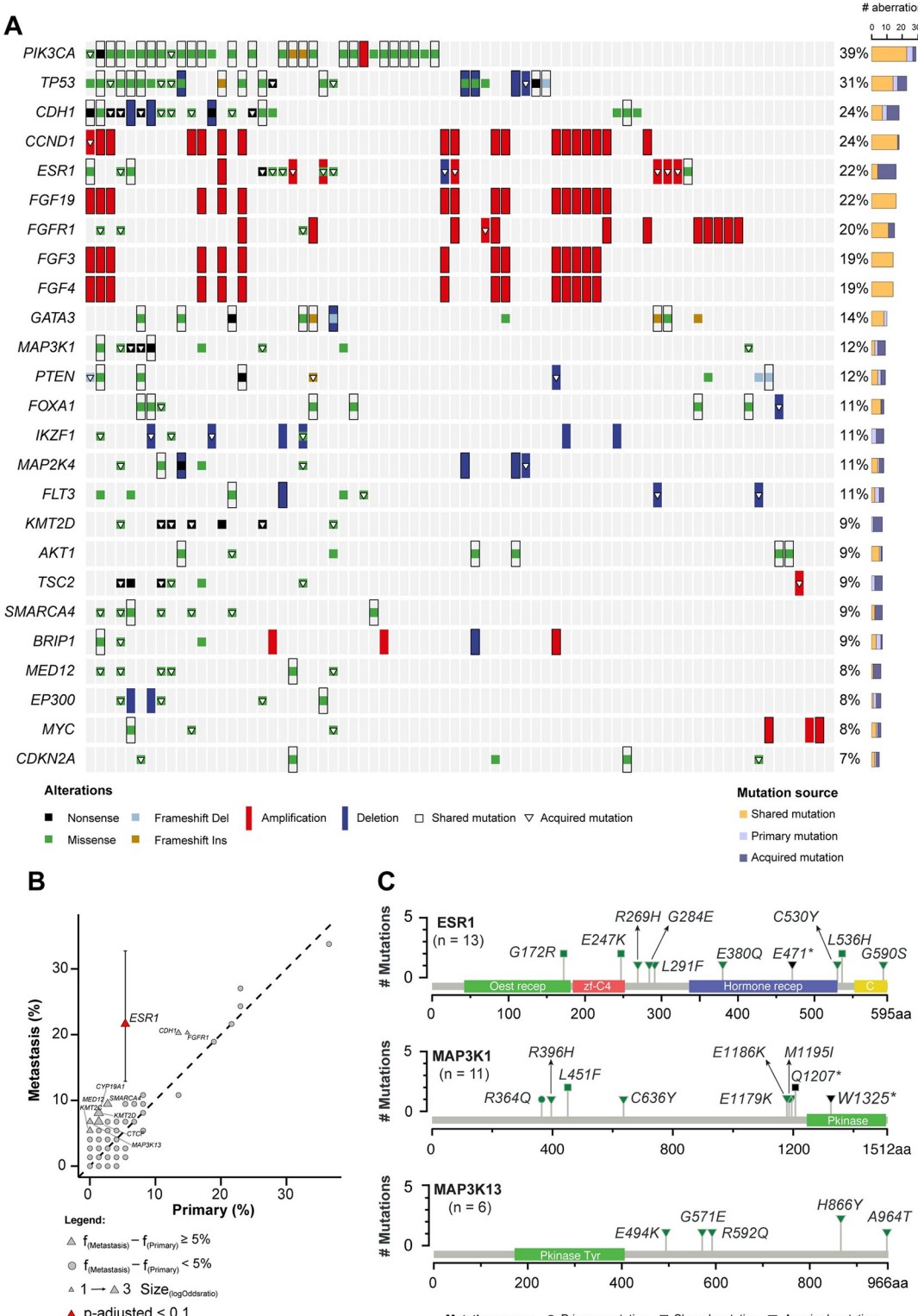

**Fig 1. Repertoire of genomic alterations in primary and metastatic HR+ HER2- breast cancer (BC).** A) Recurrent driver somatic mutations and copy number variations identified in matched primary and metastatic HR+ HER2- BC specimens (n = 74) subjected to targeted sequencing (n = 148). Cases are shown in columns, whereas genes are shown in rows. Mutation types are color-coded according to the legend. The total number of alterations detected in individual genes is displayed on the bar plot (right). For graphical purpose, only the top 25 genes are shown. B) The scatter plot reports the mutational frequencies

in matched primary (n = 74) and metastatic (n = 74) tumor samples. Color indicates statistical significance (p-value adjusted by false discovery rate ≤ 0.1), the shape of the points reflects the difference between mutational frequency in metastatic vs. primary tumor samples (triangle = frequency difference ≥ 5%, circle = frequency difference < 5%). Confidence intervals for proportions is reported for significant genes. C) Schematic representation of the protein domains of *ESR1*, *MAP3K1* and *MAP3K13* and of the somatic mutations in matched primary and metastatic HR+ HER2- BC specimens (n = 74). Mutations are color-coded according to the legend, and their overall occurrence is represented on the y-axis.

in mBC specimens (**Fig 1B**). Among acquired aberrations, only *ESR1* mutations were found to be significantly enriched in mBC specimens after adjustment for multiple testing (FDR = 0.0048, **Fig 1B**). Among genes less commonly mutated in BC, *MAP3K1* mutations were enriched in mBC (6.7%), while *MAP3K13* mutations were exclusively detected in mBC (6.7%) specimens although not reaching statistical significance after multiple testing correction due to the low number of events in out cohort (**Fig 1C**). Taken together, *MAP3K1* or *MAP3K13* acquired mutations were found in eight (10.8%) mBC specimens.

## Acquired *ESR1* sCNAs in mBC and subclonal frequency mutations in eBC

We identified eight cases with bona-fide *ESR1* sCNAs. *ESR1* high-level copy gains were detected exclusively in the metastatic samples in six cases, and in both primary and metastatic specimens in one case. Finally, one patient showed an *ESR1* copy loss in the metastatic deposit only.

Due to the novelty of *ESR1* CNVs, we sought to confirm *ESR1* gene amplification by FISH analysis in mBC cases, and to assess the average *ESR1* gene copy number in the corresponding eBC samples. To this aim, we designed probes hybridizing to *ESR1* genomic region and chr6 centromere, as well as to *MYB* genomic locus, which maps on ch6 q23.3 between the centromere and *ESR1* genomic locus. Of note, FISH analysis confirmed NGS findings, identifying both the cases with acquired *ESR1* amplification (i.e., detected in mBC but not in eBC specimens) (**Fig 2A**, first two karyograms and FISH pictures) and the case in which *ESR1* gene amplification was detected both in eBC and mBC samples (**Fig 2A**, lower karyograms and FISH pictures). Of interest, since amplicons in our TGS panel are present both at the 5' and the 3' of *ESR1*, we can conclude that *ESR1* amplification is indeed a focal and not an arm-level phenomenon. This was confirmed by FISH analysis, in which *ESR1* amplification (ratio > 2) was retained also when normalized against the *MYB* gene. In an eBC specimen, we also identified a low frequency mutation in *ESR1* (variant allele frequency—VAF = 2.6% in a sample with 3n ploidy and 40% purity estimated with ABSOLUTE), which was expanded in its matched mBC sample (VAF 8.8%, sub-3n ploidy and 43% purity, see **S4 Table**). Although this observation is limited to a single case, it supports the hypothesis that, although uncommonly, subclonal *ESR1* mutations may occur in ET-naive HR+ HER2- BC specimens, and they may expand ET-induced selective evolutive pressure.

## *ESR1* aberrations are selected upon estrogen deprivation during adjuvant treatment

Overall, we detected acquired *ESR1* mutations/amplifications in 15 cases (20%). While *ESR1* mutations are a well-established mechanism of resistance to ET, it is less clear if specific types of ETs, such as AI plus/minus LHRHa or TAM plus/minus LHRHa, are more likely to cause the selection of tumor clones bearing *ESR1* gene mutations. In addition, since acquired *ESR1* amplifications in mBC specimens are a novel finding of our study, we also investigated whether specific types of ET are associated with the acquisition of *ESR1* gene amplifications and/or the selection of tumor clones bearing *ESR1* amplification. In our cohort, 12 patients

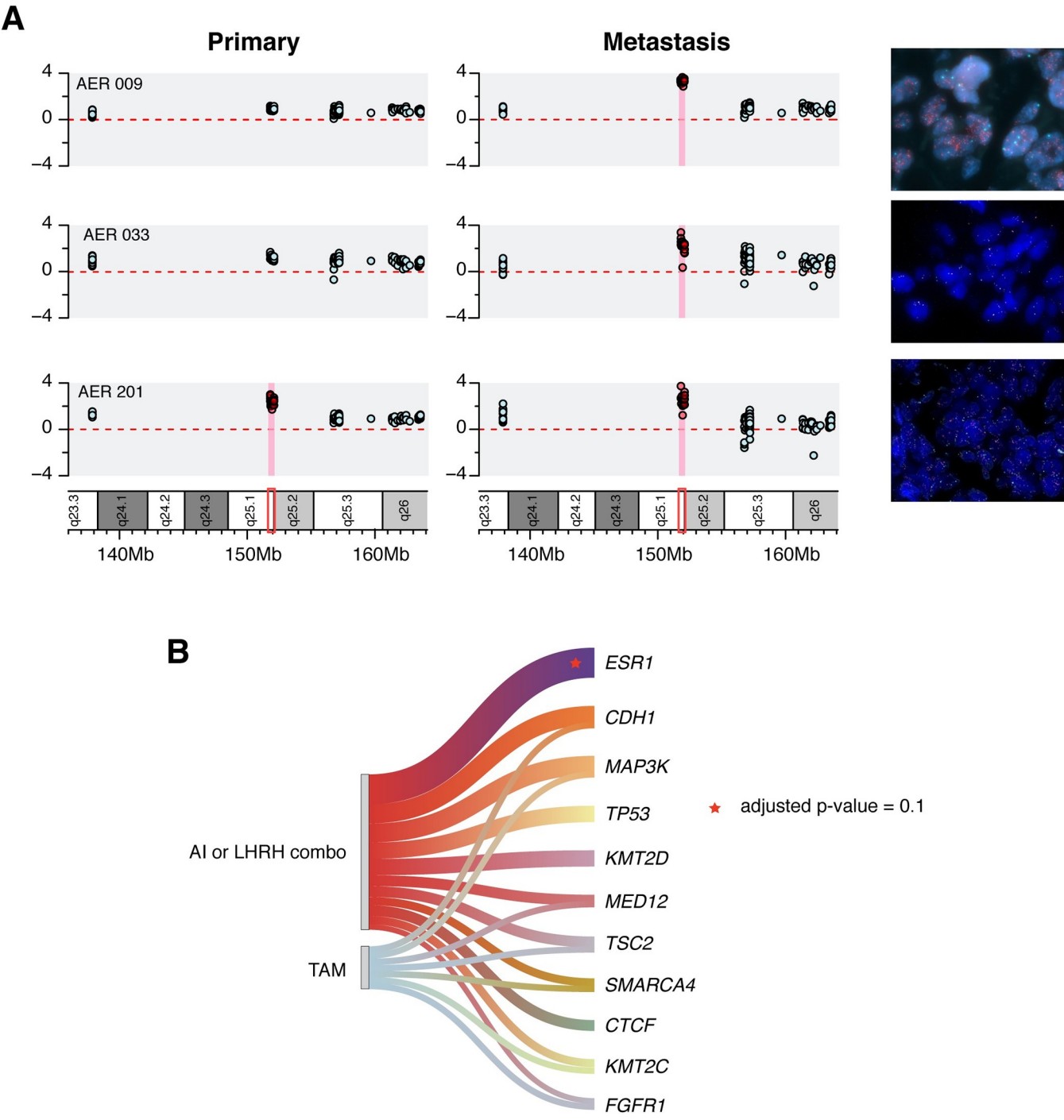

**Fig 2. *ESR1* enrichment in metastatic HR+ HER2- BC samples.** A) *ESR1* gene sCNV in three representative cases. Each row represents a case, and each box indicates the log-ratio levels for matched primary and metastatic tumor specimens. Red square shows the exact *ESR1* region. The red background highlights the amplification of the *ESR1* gene. FISH-based validation of each *ESR1* amplification is also shown (right panel). B) Recurrent genomic alterations in metastatic tumor specimens, and their association with different types of endocrine therapy (ET). ET is classified according to specific clinically relevant groups. Statistically significant associations are shown as stars (adjusted p-value = 0.1).

were treated with an AI (16%), 43 patients underwent estrogen deprivation through a LHRHa (mostly in combination with TAM, 54%), while the remaining patients received TAM alone (n = 11; 15%), or TAM followed by an AI (n = 5; 7%). When considered individually, different types of ET (TAM; AI; LHRHa plus either TAM or AI) were not associated with any specific tumor aberration (see **S2 Fig**). However, since LHRHa reduce circulating levels of estrogens in pre-menopausal women similarly to AIs in post-menopausal women, we grouped together patients treated with AIs or with LHRHa (with either TAM or AIs). Notably, *ESR1* mutations were only detected in patients undergoing estrogen-reducing therapies (n = 6; 8%), (i.e., AIs or LHRHa combinations see **Fig 2B**). Similarly, acquired *ESR1* gene amplifications were selectively detected in patients receiving AIs or LHRHa-based ET (n = 6; 8%). These associations were statistically significant with a p-value = 0.044. Together, these findings indicate that the selection of both *ESR1* mutations and *ESR1* amplifications in human HR+ HER2- BC may be driven by estrogen deprivation. Other genes whose mutations were found to be enriched, although not statistically significantly, in patients receiving estrogen deprivation adjuvant ET included *CTCF*, which encodes a transcription factor previously associated with endocrine resistance in BC [24], and *TP53*. We did not find any significant association between acquired *ESR1* alterations (mutations, amplifications), or mutations in other BC-related genes, and the administration of adjuvant chemotherapy, which occurred in 43% of patients included in our study cohort (mostly anthracycline-containing regimens).

## Acquired *MAP3K* mutations are independent predictors of worse clinical outcome in relapsing HR+ HER2- BC patients

Razavi et al. [3] previously reported on the role of the MAPK pathway alterations in mediating resistance to ET in human HR+ HER2- BC. Consistent with these data, in our study cohort *MAP3K13* mutations were detected in mBC samples (n = 5), but not in the corresponding eBC samples. To investigate the potential prognostic role of individual acquired gene mutations, we evaluated whether any acquired mutation with a difference of at least 5% in frequency between paired primary and relapsing tumors was associated with DRFS or OS. Notably, only *MAP3K* mutations (including both *MAP3K1* and *MAP3K13* mutations) were associated with significantly shorter DRFS and OS (HR = 2.91, 95%CI = 1.35–6.34, p-value = 0.0049 and HR = 2.64, 95%CI = 1.18–5.94, p-value = 0.015, respectively, see **Fig 3A and 3B**). The association between *MAP3K* mutations and worse clinical outcomes retained statistical significance after adjustment for patient menopausal status, primary tumor size (pT), nodal involvement at diagnosis (pN), tumor grade and the percentage of Ki-67 labelling index (adjusted HR for DRFS: 3.29, 95%CI = 1.46–7.42, p-value = 0.0039 and adjusted HR for OS: 5.46, 95%CI = 2.19–13.60, p-value = 0.0002 respectively, see **Fig 3C and 3D**).

## The acquisition of novel driver mutations is independently associated with worse patient prognosis

Our targeted gene panel, which was designed to encompass the most commonly altered BC-related genes, was not specifically meant to provide an accurate estimation of tumor mutational burden (TMB) according to current guidelines [25]. However, we reasoned that the overall number of acquired driver mutations (here referred to as "driver mutational load") might have prognostic relevance. Overall, we observed a high heterogeneity in the frequency of driver mutations (see **Fig 4A**). Of the 74 matched cases analyzed, 48 (65%) did not show any acquired driver mutation or sCNA in metastatic samples, and 16 (33%) of them actually lost driver mutations when compared to their matched primary tumors, possibly suggesting clonal restriction upon adjuvant treatments (**Fig 4B**). By contrast, in 26 (35%) cases, we found

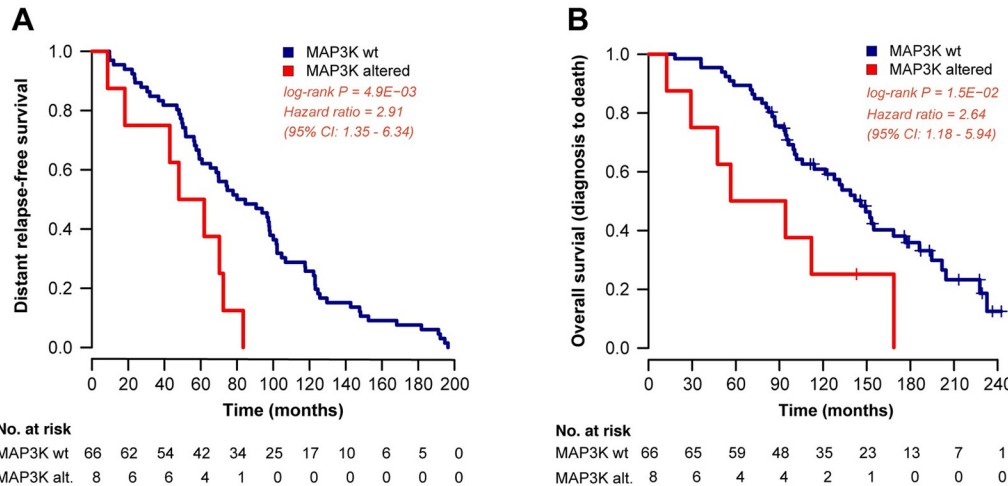

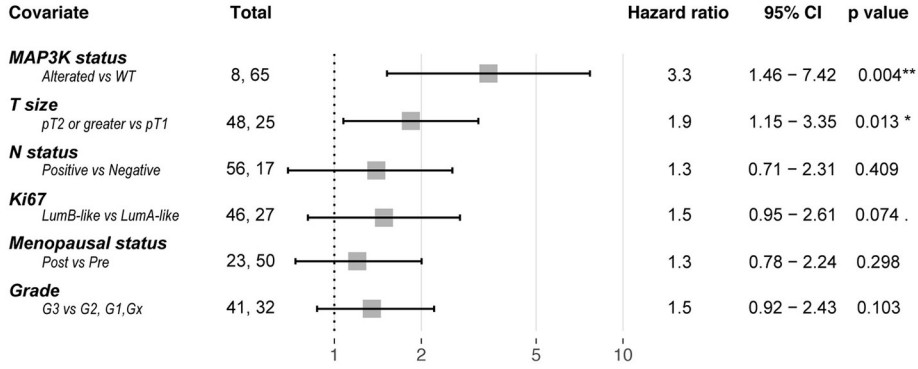

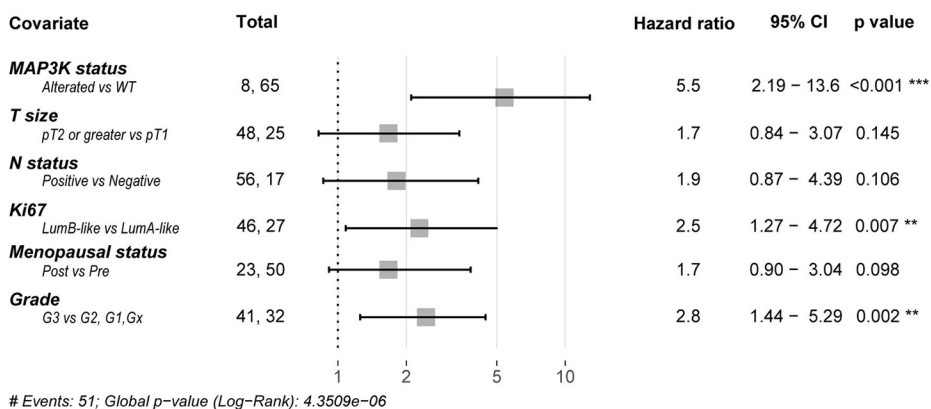

**Fig 3. Association between *MAP3K* alterations and clinical outcome.** Kaplan Meier curves displaying distant relapse-free survival (DRFS) (A) and overall survival (OS) (B) in patients with *MAP3K* gene alteration (red curves) as compared with patients with wild-type *MAP3K* status (blue curves). Forest plots indicating the hazard ratios for DRFS (C) and OS (D), and the corresponding confidence intervals, in *MAP3K*-altered *and MAP3K*-wild type patients. Multivariable Cox analysis is adjusted for tumor size, lymph node involvement, Ki67, menopausal status and tumor grade of the primary tumor.

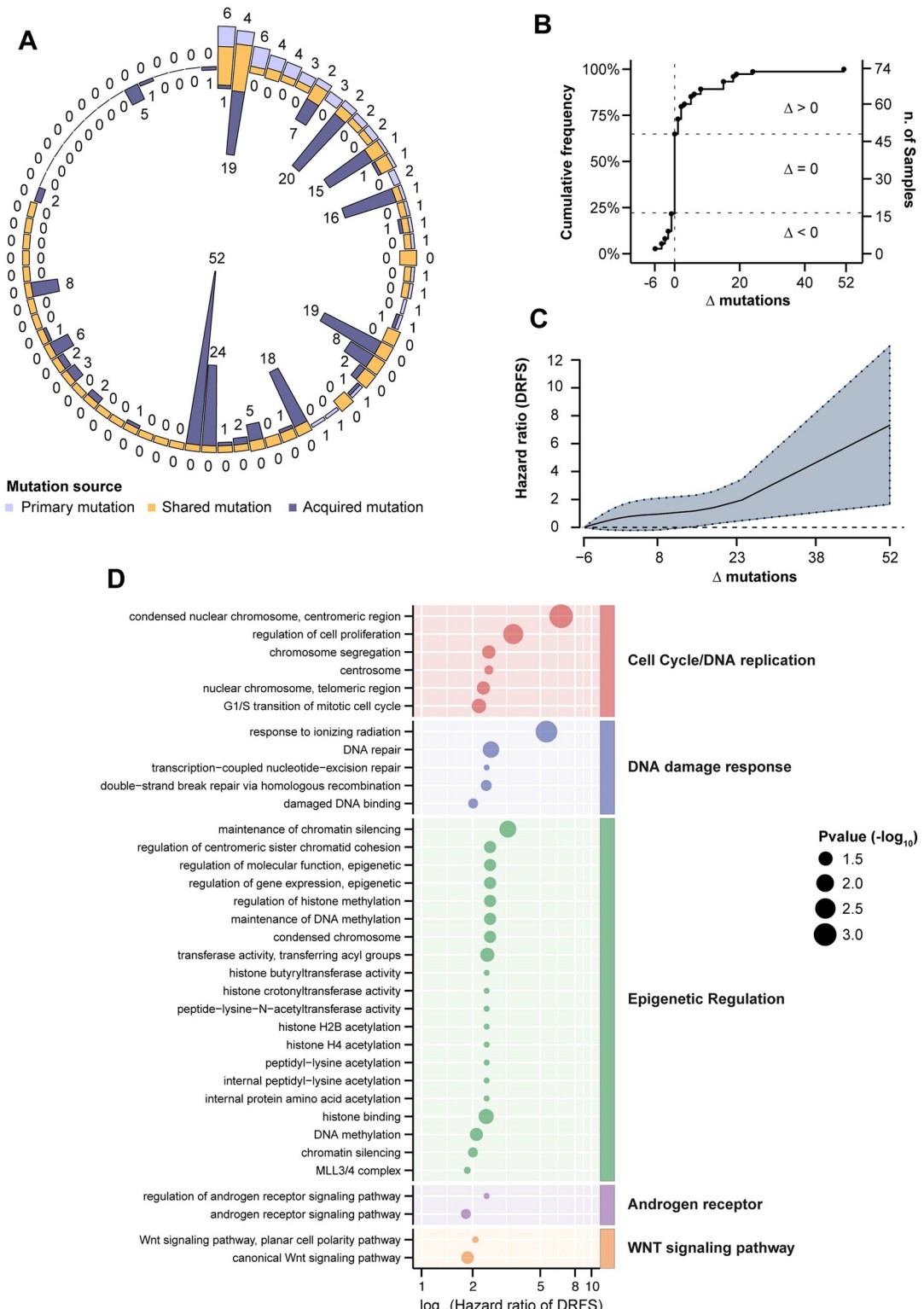

**Fig 4. Genomic spectrum of acquired driver alterations.** A) The circle graph represents for each case (n = 74) the proportion of driver mutations detected in primary and/or metastatic tumor samples. Outer numbers represent mutations of eBC, inner numbers represent mutations of mBC. B) Cumulative frequency of the difference (Δ) between number of mutations in metastatic *vs.* primary tumor samples (Δ < 0, number of driver mutations in the primary tumor greater than in the corresponding metastatic sample; Δ = 0, equal number of driver mutations in primary *vs.* metastatic tumor; Δ > 0, number of

driver mutations in the primary sample lower than in the metastatic sample. C) Non-linear relationship between the difference of driver mutations in metastasis/primary pair (Δ, x-axis), and DRFS hazard ratio of Schoenfeld residuals (y-axis). The analysis is adjusted for T/N status, Ki67, menopausal status and tumor grade. The solid line represents a penalized spline fit of the predicting variables, while the dashed lines show 95% confidence intervals. D) Functional analysis of Gene Ontology (GO) terms associated to cell cycle, DDR, epigenetic regulation, androgen receptor activity and WNT signaling pathway. The size of the dots is inversely proportional to the p values of estimated hazard ratio (x-axis) displayed in $\log_{10}$ scale. P values are reported in S5 Table.

oncogenic driver mutations or sCNA in metastatic samples that were not detected in matched eBC specimens. Interestingly, primary tumors with a higher number of baseline driver mutations were more likely to acquire a higher number of novel mutations in their matched metastatic samples (rho = 0.39 and p-value = 0.0004). The gain of novel driver mutations was associated with significantly worse DRFS in a way that was almost linearly proportional to their number, and independent of relevant clinico-pathological variables (for Δ mutations = 52, HR = 7.32, 95% CI = 1.65–12.98 and p-value = 0.0062, see **Fig 4C**). With regards to the biological pathways potentially affected by acquired driver aberrations in mBC, we identified terms related to DNA damage repair and DNA replication, epigenetic reprogramming (acetylation and methylation transferase activity, histone-related terms), androgen receptor and Wnt pathways. These terms were significantly enriched in tumor specimens characterized by a higher number of acquired driver mutations and associated with significantly worse patient DRFS (see **Fig 4D** and **S5 Table**).

## Discussion

The main aim of our study was to investigate the evolution of the mutational profile of HR + HER2- BC relapsing during or after the completion of adjuvant ET following surgical resection, with or without (neo)adjuvant chemotherapy. We therefore compared tumor genomic alterations in matched mBC and eBC tumor specimens. Although our methodology hindered a computational analysis of genomic evolution based on mathematical models, the relative prevalence of driver gene aberrations observed in relapsing tumors allowed us to pinpoint mechanisms of pathway reactivation and by-pass potentially leading to ET resistance. Frequencies of the most common genomic alterations reported in this study are consistent with results of previously published analyses in larger HR+ HER2- BC patient series [3,26,27] thus supporting the reliability of our methodology and the representativeness of our clinical cohort. To our knowledge, this is the first systematical investigation of *ESR1* CNVs by NGS in primary and metastatic BC samples, followed by transversal validation with triple color FISH. Previous data on *ESR1* amplification are scarce [28,29], based on heterogeneous techniques which were not properly validated. We found that *ESR1* gene amplification was specifically selected in patients treated with adjuvant AI- and/or LHRHa-based therapy, thus suggesting that it could be a novel mechanism of tumor resistance to ET in HR+ HER2- BC beyond the more studied and previously characterized *ESR1* mutations [5,6,30–33]. Consistent with this hypothesis, *ESR1* gene amplification was almost exclusively detected in relapsing tumors (85.7%) and it was not concomitant to *ESR1* gene mutation in all cases but one. Collectively, our data suggest that ESR1 pathway reactivation by gene mutation or amplification may account for tumor resistance and relapse in roughly 15% of HR+ HER- BC patients.

Preclinical mechanistic experiments are needed to dissect the functional relevance of ESR1 amplifications more clearly in HR+ HER2- BC cells exposed to different types of pharmacological treatments, including *in vitro* estrogen deprivation (which mimics the *in vivo* effects of AIs) +/- CDK 4/6 inhibitors or everolimus, fulvestrant +/- CDK 4/6 inhibitors, or fulvestrant +/- alpelisib (in *PIK3CA*-mutated cancers). Future studies in larger clinical cohorts are needed

to confirm the functional significance and clinical consequences of *ESR1* amplification in both primary and metastatic HR+ HER2- BC specimens, and to investigate its clinical relevance in terms of tumor sensitivity/resistance to standard first-line therapies, i.e., CDK4/6 inhibitors in combination with AIs or fulvestrant [34–39], in the treatment of advanced BC patients. Interestingly, new SERDs under clinical development are effective in inhibiting constitutively active ERα that is encoded by mutated *ESR1* gene [40]. In this perspective, it will be interesting to evaluate if these new SERDs are also effective against HR+ HER2- BC cells bearing *ESR1* gene amplification.

In our study, the occurrence of *MAP3K* mutations, emerging either during or after the completion of adjuvant ET, was the only covariate significantly associated with worse prognosis in a multivariable Cox model including clinically relevant patient- and tumor-related variables, such as menopausal status, T and N stage, tumor grade and Ki-67 labelling index. Within the established crosstalk between ERα and MAPK signaling axes, constitutive activation of the MAPK pathway could bypass ERα inhibition, eventually promoting tumor cell growth, proliferation and resistance to ETs [10,41,42]. Despite the low number of patients included in this study, our findings indicate that *MAP3K* mutations may be a novel biomarker of shorter HR+ HER2- BC patient DRFS/OS.

In this study, we also found that an enrichment of genomic alterations in BC-related genes, a proxy of the mutational tumor load, was associated with significantly worse DRFS. Interestingly, most of the acquired genomic alterations occurred in genes encoding DNA repair enzymes or epigenetic regulators, which may be associated with an hypermutated phenotype potentially leading to therapy resistance.

The experimental design adopted in our study was not aimed at recapitulating the clonal evolution upon the selective pressure of systemic therapy. Therefore, we cannot exclude in principle that at least a fraction of the resistance mutations found exclusively in metastatic samples could be overlooked in the primary tumor due to a very low frequency or spatial heterogeneity. Along this line, we reported a case harboring an ESR1 mutation, in which the variant allele frequency raised from 2,6% in the primary tumor sample to 8,8% in the metastatic sample, letting us speculate that this could represent an example of clonal expansion upon therapy selective pressure.

Main limitations of our study are: 1) the relatively low number of patients included in our clinical cohort, which did not allow us to reliably analyze genomic alterations occurring at low frequency ($< 5\%$). However, to the best of our knowledge this is one of the largest clinically annotated cohorts in which tumor genomic profiles were analyzed in matched eBC and mBC specimens; 2) the use of a customized genomic panel that, although including the most common genes and hotspot alterations previously detected in BC patients, could potentially limit the range of detectable alterations; 3) the relative heterogeneity in (neo)adjuvant chemotherapy and ET regimens delivered to patients.

In conclusion, *ESR1* amplifications and *MAP3K* mutations are novel genomic alterations that are selected upon adjuvant ETs in patients with advanced HR+ HER2- BC, and which represent promising targets of pharmacologic inhibition to reverse acquired tumor resistance. *MAP3K* mutations and a global increase in the number of gene aberrations were also associated with worse patient DRFS and OS, and they could be useful in predicting clinical outcomes in patients with metastatic HR+ HER2- BC.

## Supporting information

**S1 Fig. (CONSORT-like flow chart).**
(PNG)

**S2 Fig. (Associations between gene alteration and ET, broad classes).**
(TIF)

**S1 Table. (Detailed Patient-Level Clinical and Treatment Data for the Study Cohort).**
(XLSX)

**S2 Table. (Sequencing Gene Content, Gene Targets and BED file of the design).**
(XLSX)

**S3 Table. (Cohort overview).**
(XLSX)

**S4 Table. (Purity and ploidy estimate).**
(XLSX)

**S5 Table. (GO terms).**
(XLSX)

**S1 Star Methods. (Supplementary methods).**
(DOCX)

## Acknowledgments

GZ wishes to thank Dr. P. Blandini MD for his precious insights in the final draft of the manuscript.

## Author Contributions

**Conceptualization:** Lorenzo Ferrando, Andrea Vingiani, Anna Garuti, Marco Colleoni, Giuseppe Viale, Gabriele Zoppoli, Giancarlo Pruneri.

**Data curation:** Lorenzo Ferrando, Andrea Vingiani, Anna Garuti, Claudio Vernieri, Antonino Belfiore, Luca Agnelli, Gianpaolo Dagrada, Diana Ivanoiu, Giuseppina Bonizzi, Martina Dameri, Francesco Ravera, Marco Colleoni, Giuseppe Viale, Luca Magnani, Alberto Ballestrero, Gabriele Zoppoli, Giancarlo Pruneri.

**Formal analysis:** Lorenzo Ferrando, Andrea Vingiani, Anna Garuti, Claudio Vernieri, Antonino Belfiore, Luca Agnelli, Gianpaolo Dagrada, Diana Ivanoiu, Giuseppina Bonizzi, Martina Dameri, Francesco Ravera, Luca Magnani, Gabriele Zoppoli, Giancarlo Pruneri.

**Funding acquisition:** Giuseppina Bonizzi, Gabriele Zoppoli, Giancarlo Pruneri.

**Investigation:** Lorenzo Ferrando, Andrea Vingiani, Anna Garuti, Claudio Vernieri, Antonino Belfiore, Luca Agnelli, Gianpaolo Dagrada, Giuseppina Bonizzi, Elisabetta Munzone, Luana Lippolis, Martina Dameri, Francesco Ravera, Marco Colleoni, Giuseppe Viale, Luca Magnani, Alberto Ballestrero, Gabriele Zoppoli, Giancarlo Pruneri.

**Methodology:** Lorenzo Ferrando, Andrea Vingiani, Anna Garuti, Antonino Belfiore, Luca Agnelli, Gianpaolo Dagrada, Diana Ivanoiu, Giuseppina Bonizzi, Elisabetta Munzone, Luana Lippolis, Marco Colleoni, Giuseppe Viale, Luca Magnani, Gabriele Zoppoli, Giancarlo Pruneri.

**Project administration:** Lorenzo Ferrando, Andrea Vingiani, Elisabetta Munzone, Giuseppe Viale, Giancarlo Pruneri.

**Resources:** Anna Garuti, Luca Agnelli, Luana Lippolis, Gabriele Zoppoli.

**Software:** Lorenzo Ferrando, Luca Agnelli, Luca Magnani, Gabriele Zoppoli.

**Supervision:** Lorenzo Ferrando, Andrea Vingiani, Anna Garuti, Claudio Vernieri, Elisabetta Munzone, Marco Colleoni, Giuseppe Viale, Luca Magnani, Alberto Ballestrero, Gabriele Zoppoli, Giancarlo Pruneri.

**Validation:** Lorenzo Ferrando, Anna Garuti, Claudio Vernieri, Elisabetta Munzone, Luana Lippolis, Marco Colleoni, Alberto Ballestrero, Gabriele Zoppoli, Giancarlo Pruneri.

**Visualization:** Lorenzo Ferrando, Luana Lippolis, Marco Colleoni, Luca Magnani, Gabriele Zoppoli, Giancarlo Pruneri.

**Writing – original draft:** Lorenzo Ferrando, Andrea Vingiani, Anna Garuti, Claudio Vernieri, Antonino Belfiore, Luca Agnelli, Gianpaolo Dagrada, Diana Ivanoiu, Giuseppina Bonizzi, Elisabetta Munzone, Martina Dameri, Francesco Ravera, Marco Colleoni, Giuseppe Viale, Luca Magnani, Alberto Ballestrero, Gabriele Zoppoli, Giancarlo Pruneri.

**Writing – review & editing:** Lorenzo Ferrando, Andrea Vingiani, Anna Garuti, Claudio Vernieri, Antonino Belfiore, Luca Agnelli, Gianpaolo Dagrada, Diana Ivanoiu, Giuseppina Bonizzi, Elisabetta Munzone, Martina Dameri, Francesco Ravera, Marco Colleoni, Giuseppe Viale, Luca Magnani, Alberto Ballestrero, Gabriele Zoppoli, Giancarlo Pruneri.

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
