## [Decision Letter · Decision Letter 0]

27 Sep 2022

Dear Dr Pruneri,

Thank you very much for submitting your Research Article entitled 'ESR1 gene amplification and MAP3K mutations are selected during adjuvant endocrine therapies in relapsing Hormone Receptor-positive, HER2-negative breast cancer (HR+ HER2- BC)' to PLOS Genetics.

The manuscript was fully evaluated at the editorial level and by independent peer reviewers. The reviewers appreciated the attention to an important topic but identified some concerns that we ask you address in a revised manuscript.

We therefore ask you to modify the manuscript according to the review recommendations. Your revisions should address the specific points made by each reviewer.

[LINK]

Yours sincerely,

David J. Kwiatkowski

Section Editor

PLOS Genetics

David Kwiatkowski

Section Editor

PLOS Genetics

Reviewer's Responses to Questions

**Comments to the Authors:**

Reviewer #1: The authors of this article performed a genomic characterization of 74 surgically resected HR+ HER2- BC patients relapsing during or at the completion of adjuvant ET. They compared tumor genomic alterations in matched mBC and eBC tumor specimens: along with previously reported genomic alterations, including PIK3CA, TP53, CDH1, GATA3 and ESR1 mutations/deletions, they found that ESR1 gene amplification (confirmed by FISH) and MAP3K mutations were enriched in metastatic lesions as compared to matched primary tumors of patients treated with either adjuvant aromatase inhibitors or LHRH analogs + tamoxifen. Patients with tumor bearing MAP3K mutations in metastatic lesions had significantly worse survival in multivariate analysis.

The manuscript is well written and study results are clearly reported and discussed.

Some comments/suggestions:

1. In Methods section, the authors state that study patients were selected based on the availability of clinical data regarding “adjuvant ET (if any)”. If adjuvant ET was mandatory and all study patients received adjuvant ET, please eliminate “(if any)”.

2. Among patients characteristics, the authors should report how many patients relapsed during adjuvant ET and how many had disease relapse after the completion of adjuvant ET.

3. The authors report that 12 patients were treated with AI, 3 with LHRH + AI, 5 with Tam followed by AI, 11 with Tam alone, 40 with LHRH + Tam, and 3 patients were treated in random clinical trials (total: 74). Furthermore, they write (page 8, line 1) that in 2 patients, they were unable to retrieve information about the type of ET prescribed. Please clarify to which patients they refer.

4. Among acquired aberrations detected by NGS, only ESR1 mutations were found to be significantly enriched in mBC specimens after adjustment for multiple testing. If so, the authors should better clarify that the observed enrichment of MAP3K mutations in metastatic lesions was not statistically significant after multiple testing.

5. It would be interesting to know DRFS and OS of study patients by PIK3CA mutation status. PIK3CA and ESR1 mutation status should also be included as covariates in multivariable Cox analysis.

Reviewer #2: This study by Lorenzo et al. presents genomic analysis of ER+ breast tumors that recurred after endocrine therapy. Resistance to endocrine therapy is a significant problem in ER+ breast cancer and the question is important. Strengths of the study include matched primary tumor, germline DNA, and recurrent tissue for direct comparison of acquired features. This study additionally adds granular analysis of how patients were treated. Study is limited by relatively small sample size compared to prior works (Razavi) and lack of mechanistic or translational studies. Additionally, the two main alterations described, ESR1 and MAPK are well described.

- Table 1 describes luminal A like and Luminal B-like based on Ki-67 – why was a more established transcriptomic definition not utilized?

- Figure 2 B is confusing to me. It seems that the treatment should be on the left because it occurs temporally before the alterations. The two diagrams also present largely overlapping data and I would encourage the authors to pick one or clearly highlight the value of both.

- The ESR1 by treatment type sensitivity analysis is based on a very small number of patients and strength of conclusions (ESR1 mutations and amplifications is driven by estrogen deprivation) should be toned down.

- Another explanation is that rather than driving mutagenesis (which is possible), that low abundance subpopulations harbor mutations and then are selected by therapy. Thus mutations may confer differential sensitivity to treatment which then manifest as different growth rates under selective pressures. This alternative hypothesis should be discussed.

**Have all data underlying the figures and results presented in the manuscript been provided?**

Reviewer #1: Yes

Reviewer #2: Yes

PLOS authors have the option to publish the peer review history of their article (what does this mean?). If published, this will include your full peer review and any attached files.

Reviewer #1: No

Reviewer #2: No

---

## [Decision Letter · Decision Letter 1]

8 Dec 2022

Dear Dr Pruneri,

We are pleased to inform you that your manuscript entitled "ESR1 gene amplification and MAP3K mutations are selected during adjuvant endocrine therapies in relapsing Hormone Receptor-positive, HER2-negative breast cancer (HR+ HER2- BC)" has been editorially accepted for publication in PLOS Genetics. Congratulations!

Yours sincerely,

David J. Kwiatkowski

Section Editor

PLOS Genetics

David Kwiatkowski

Section Editor

PLOS Genetics

Comments from the reviewers (if applicable):

Reviewer's Responses to Questions

**Comments to the Authors:**

Reviewer #1: The authors have satisfactorily addressed all my comments and requests.

Reviewer #2: Reviews have been adequately addressed.

**Have all data underlying the figures and results presented in the manuscript been provided?**

Reviewer #1: Yes

Reviewer #2: Yes

PLOS authors have the option to publish the peer review history of their article (what does this mean?). If published, this will include your full peer review and any attached files.

Reviewer #1: No

Reviewer #2: No

**Data Deposition**

http://datadryad.org/submit?journalID=pgenetics&manu=PGENETICS-D-22-00807R1

**Press Queries**

---

## [Editor Report · Acceptance letter]

28 Dec 2022

PGENETICS-D-22-00807R1 

ESR1 gene amplification and MAP3K mutations are selected during adjuvant endocrine therapies in relapsing Hormone Receptor-positive, HER2-negative breast cancer (HR+ HER2- BC) 

Dear Dr Pruneri, 

We are pleased to inform you that your manuscript entitled "ESR1 gene amplification and MAP3K mutations are selected during adjuvant endocrine therapies in relapsing Hormone Receptor-positive, HER2-negative breast cancer (HR+ HER2- BC)" has been formally accepted for publication in PLOS Genetics! Your manuscript is now with our production department and you will be notified of the publication date in due course.

With kind regards,

Anita Estes

PLOS Genetics

On behalf of:
